# Machine learning assisted vector atomic magnetometry

Xin Meng[1], Youwei Zhang[1], Xichang Zhang [1], Shenchao Jin[1], Tingran Wang[2], Liang Jiang [3], Liantuan Xiao[4,5], Suotang Jia [4,5] & Yanhong Xiao [4,5] ✉

Multiparameter sensing such as vector magnetometry often involves complex setups due to various external fields needed in explicitly connecting one measured signal to one parameter. Here, we propose a paradigm of indirect encoding for vector atomic magnetometry based on machine learning. We encode the three-dimensional magnetic-field information in the set of four simultaneously acquired signals associated with the optical rotation of a laser beam traversing the atomic sample. The map between the recorded signals and the vectorial field information is established through a pre-trained deep neural network. We demonstrate experimentally a single-shot all optical vector atomic magnetometer, with a simple scalar-magnetometer design employing only one elliptically-polarized laser beam and no additional coils. Magnetic field amplitude sensitivities of about $100\,\text{fT}/\sqrt{\text{Hz}}$ and angular sensitivities of about $100 \sim 200\,\mu rad/\sqrt{\text{Hz}}$ (for a magnetic field of around 140 nT) are derived from the neural network. Our approach can reduce the complexity of the architecture of vector magnetometers, and may shed light on the general design of multiparameter sensing.

Developing atomic sensors with high sensitivity and compact configuration is a topic of great interest in quantum science and technologies. Prominent measurement devices including atomic clocks[1,2], atom interferometers[3], magnetometers[4] and microwave sensors[5] etc., are under active pursuit and play important roles in both fundamental research and real-life applications ranging from new physics search[6] to navigation and medical diagnosis[7,8]. While in most scenarios the sensing process can be described by a single parameter estimation problem, multiparameter estimation[9,10] has recently attracted attention both theoretically and experimentally. Notable examples are measurements of a multi-dimensional field, identification of a spatial structure[11] or multi-frequency signals[12]. In general, multiparameter measurement requires a more involved sensor architecture, such as applying several electromagnetic fields along different directions to interact with the atoms, or performing successive interrogations under varied conditions. Furthermore, the relation between the observable readings and the parameters can be complex and decoding may require model fitting or elaborate data analysis techniques[13–15].

Machine learning (ML), as a part of artificial intelligence, involves model-building based on sample data, or training data, to "learn" and then to make predictions without an explicit programme. ML is used widely for instance in speech recognition[16], computer vision[17], social network filtering[18], medical diagnosis[19,20] etc. Recently, ML has been applied in many fields of physics, to name a few, ultrafast laser science[21,22], ultracold atoms[23], many-body physics[24], classification of quantum phases[25], and quantum error correction[26]. Some works have also demonstrated its use in atomic sensors[12,27], where it was shown that ML can perform better than a physics model. However, in these proof-of-principle experiments on atomic sensors, ML is merely used in analyzing the signal's time trace to extract several frequency

[1]Department of Physics, State Key Laboratory of Surface Physics and Key Laboratory of Micro and Nano Photonic Structures (Ministry of Education), Fudan University, Shanghai 200433, China. [2]Department of Physics, The University of Chicago, Chicago, IL 60637, USA. [3]Pritzker School of Molecular Engineering, The University of Chicago, Chicago, IL 60637, USA. [4]State Key Laboratory of Quantum Optics and Quantum Optics Devices, Institute of Laser Spectroscopy, Shanxi University, Taiyuan 030006, China. [5]Collaborative Innovation Center of Extreme Optics, Shanxi University, Taiyuan 030006, China. ✉e-mail: yxiao@fudan.edu.cn

components. The potential of ML in atomic sensors, especially in multiparameter estimation, is yet to be unveiled. How to obtain the measurement sensitivity from the ML, and whether incorporating ML can significantly reduce the complexity in the sensor's hardware remains elusive.

As an example of multiparameter atomic sensor, the vector magnetometer undergoes intense investigations for it provides more complete information than its scalar counterpart and has applications in biosciences, geophysics etc. To attain the magnetic field's orientation, the sensor needs to incorporate certain axial references, for example field compensation coils[28], radio frequency fields[29-31], multiple crossing laser beams[32-36], which all inevitably complicates the setup. Also, in many schemes the three-dimensional information is obtained successively[29,37], or through sweeping the atomic resonance spectra[38-40], which may not be suitable for relatively fast or real-time field measurement[36]. Simultaneous acquisition of the three-dimensional information can be achieved by modulating bias magnetic fields in three perpendicular directions at different frequencies[41-43], thus discerning the three orthogonal magnetic field components. An all optical version of this method has been demonstrated by replacing the bias magnetic fields with orthogonally propagating laser fields imposing AC-stark shifts to the atoms[44]. However, in scenarios requiring miniaturization and high density packing of the sensors, all optical single-beam single-shot (within the sensor's response time) vector magnetometry is desired, whereas to the best of our knowledge has not been reported.

Here, we propose a paradigm for vector magnetometry based on machine learning, which enables a single-shot single-beam all optical vector magnetometer. The information is encoded in the AC components of the optical rotation signal, where the complicated and nonlinear relation between the set of four simultaneously recorded signals and the three parameters of the **B** field is established via machine learning. Removing the demand of the correspondence between one signal and one parameter as needed in most existing designs allows great simplification of the sensor structure, empowering vector magnetometry with a scalar magnetometer architecture. We further develop techniques for extracting sensitivities and frequency response of the ML-based magnetometer. The achieved sensitivities are about $100\,fT/\sqrt{Hz}$ for the field magnitude, and about $100 \sim 200\,\mu rad/\sqrt{Hz}$ for the field direction, in a room temperature Rb vapor cell. This magnetometer approach may provide insight in designing compact sensors with multiple measurement capabilities.

## Results
### Principle
Our magnetometer scheme is based on the well known nonlinear magneto-optical rotation (NMOR) process[45-48]. An elliptically polarized and frequency modulated laser beam serves as both the pump and probe field. The ellipticity of the light is optimized for balanced sensitivities of the magnetic field along different directions[49] (see Supplementary Note 3). The modulation frequency $\omega_m$ is set near the Larmor frequency of the atom $\Omega_L = \gamma B$ where $B$ is the amplitude of the total magnetic field to be measured and $\gamma$ is the gyromagnetic ratio. With the direction of **B** set as the quantization axis, the atomic levels then couple with the $\sigma^+, \sigma^-$ and $\pi$ polarization components whose amplitudes and phases depend on the orientation of the magnetic field with respect to the wave vector of the laser[38]. These optical fields and their frequency sidebands form multiple sets of Λ-type electromagnetically-induced-transparency (EIT) interactions that interfere with each other, as shown in Fig. 1a, giving rise to optical rotation effects. Since the NMOR resonance occurs when $\Omega_L$ and $\omega_m$ coincide, the phases and intensities of the transmitted sidebands naturally encode both the amplitude and the orientation of **B**. The AC components of the polarization rotation signals, i.e., the Stokes component $S_y$, are acquired by phase-sensitive detection through

frequency demodulation, where the in-phase and quadrature signals at the first and second harmonics of $\omega_m$, denoted as $X_{1,2}$ and $Y_{1,2}$, are recorded. Simultaneous recording of these four signals allows for single-shot vector magnetometry, lifting the requirement of sweeping the EIT spectrum as in refs. 38–40.

To extract the vectorial information of the magnetic field from the rotation signals, we adopt an artificial Neural Network (ANN) which is a typical algorithm of ML. By mimicking the way biological neural network learns from experience, the ANN establishes a map between input signals and output results using pre-collected data, and can thus give predictions on unknown parameters, for example, here, on the direction and magnitude of an unknown **B**. The network weights (parameters) are updated using the gradient descent algorithm[50] to minimize the defined loss function over the training data set. Each time when the NN goes through the whole training data set and returns new weights in the network is called an epoch. The loss decreases as the epoch number increases and the map is eventually established. In our scheme, the demodulated optical rotation signals $X_{1,2}$ and $Y_{1,2}$ are first collected for a range of field amplitudes and directions, and then are used to train the NN. In the end, an accurate map is established between the signal set $(X_1, Y_1, X_2, Y_2)$ and the parameter set $(B, \theta, \varphi)$, i.e., the three-dimensional field information. Here $\theta$ is normally defined as the angle between **B** and the wave vector **k** of the laser, and $\varphi$ is the azimuthal angle in the plane perpendicular to the wave vector with $\varphi = 0$ being the horizontal $x$ direction associated with the polarization axis of the optics (Fig. 1).

### Experimental setup
As shown in Fig. 1b, the light beam from an external cavity diode laser (ECDL) is near resonant with the $^{87}$Rb $D_1$ line $F = 2 \to F' = 1$ transition with 200 MHz red detuning to maximize the NMOR resonance amplitude[51,52]. The laser is frequency modulated (FM) at $\omega_m = 997$ Hz with a modulation range of 400 MHz (or modulation amplitude of 200 MHz), and its center frequency is locked via the dichroic atomic vapor laser lock[53]. The laser beam (about 2 mm in diameter) has its power (about 20 μW) stabilized in order to suppress the residual amplitude modulation. We adjust the laser polarization from linear to elliptical through two wave plates, before a cylindrical atomic vapor cell (2 cm in diameter and 7.1 cm in length) filled with enriched $^{87}$Rb at room temperature (-22 °C).

The alkene coating[54] on the inner wall of the vapor cell ensures that atoms undergo thousands of wall collisions with little destruction of their internal quantum states. The cell resides within a four-layer $\mu$-metal magnetic shield (residual field inhomogeneity in the cell is about 1 nT), together with three orthogonal sets of well-calibrated Helmholtz coils to generate the to-be-measured field **B**, with a fractional magnetic field inhomogeneity of 8/1000 within the cell. The NMOR resonance used for the magnetometer has an extracted zero-power linewidth (full width at half maximum, FWHM) of about 1 Hz, and a power broadened FWHM of about 16 Hz at the magnetometer's operational laser power 20 μW.

The Stokes component $S_y$ of the transmitted laser beam, after traversing a half-wave plate and a polarization beam splitter, is detected by a balanced photodetector in a homodyne configuration, whose output is sent to a lock-in amplifier for demodulation at frequencies $\omega_m$ and $2\omega_m$.

### Experiment results
Before collecting data for NN training, it is necessary to calibrate the residual magnetic field within the shields and the three sets of coils, in order to generate a field **B** with arbitrary direction. For a single set of coil, one can observe a good linear relation between the current applied and the magnetic field generated, but for the vector compositions of the magnetic field, the small non-orthogonality between the coils can't be neglected. Thanks to the fact that the NMOR resonance

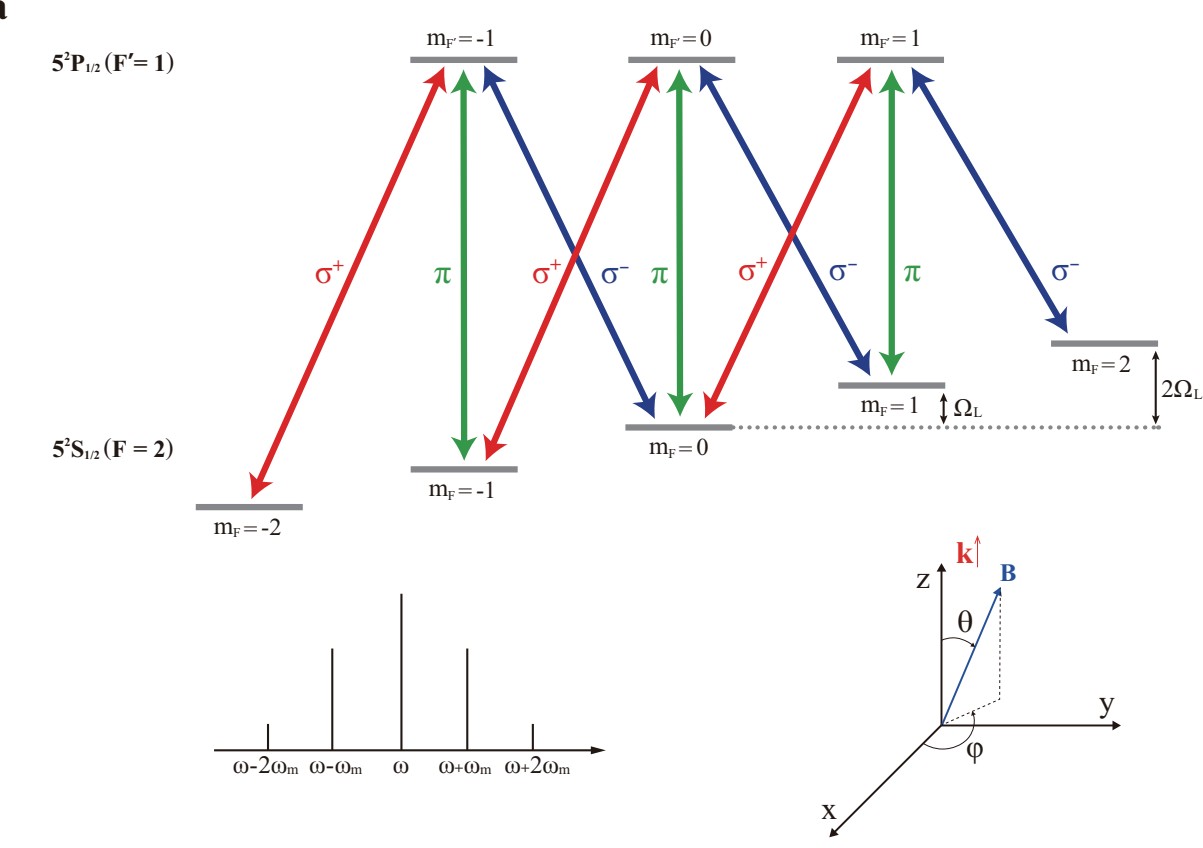

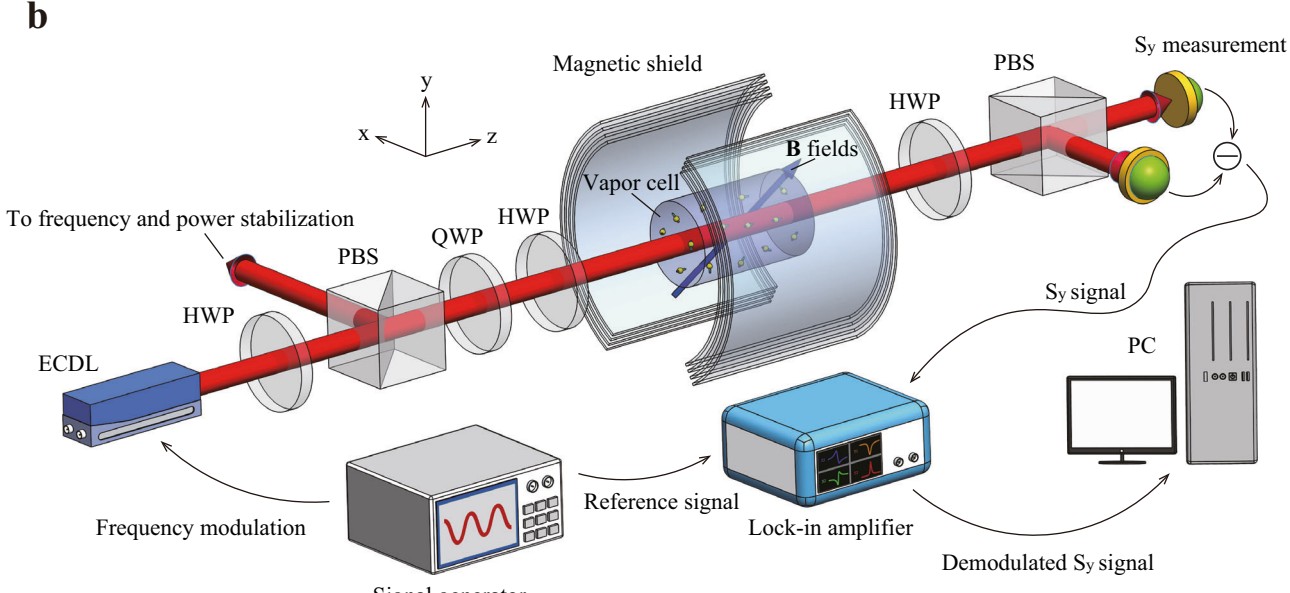

**Fig. 1 | Working principle and schematics of the single-shot all optical vector magnetometer. a** Frequency modulated elliptically polarized light interacts with the [87]Rb atom, coupling the ground state $5\ ^2S_{1/2}(F=2)$ and the excited state $5\ ^2P_{1/2}\ (F'=1)$. With the direction of the total magnetic field set as the quantization axis, atomic levels exhibit Zeeman splitting. Frequency modulation of the laser gives rise to frequency sidebands with the intervals of the modulation frequency $\omega_m$ near the Larmor frequency $\Omega_L$. The $\sigma^+, \sigma^-, \pi$ components of the laser form multiple sets of EIT. **b** Schematics of the experiment setup. ECDL external cavity diode laser, HWP half-wave plate, QWP quarter-wave plate, PBS polarization beam splitter, PC personal computer.

appears when the Larmor frequency $\Omega_L$ equals the modulation frequency $\omega_m$ or $\frac{1}{2}\omega_m$[46], these imperfections can be well calibrated. The details of the calibration process are described in Methods and Supplementary Note 2.

First, we show the observed AC optical rotation signals in the form of NMOR resonance spectra at a tilted magnetic field direction. For instance, at $\theta = 60°$, $\varphi = 60°$, when we scan the magnitude of **B**, as shown in Fig. 2a, both the first harmonic and second harmonic NMOR

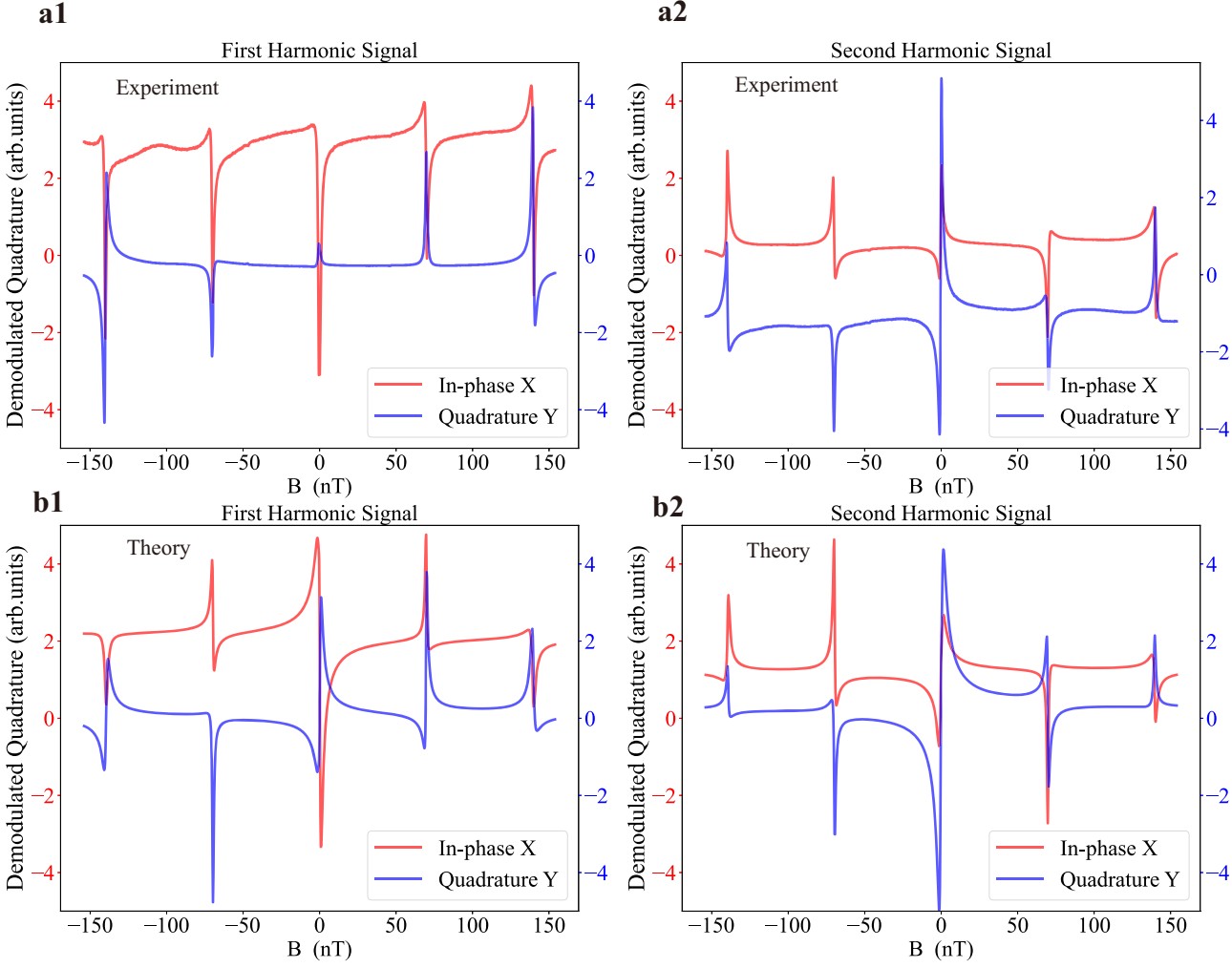

**Fig. 2 | AC quadratures of nonlinear magneto-optical rotation (NMOR) signals as a function of the magnetic field amplitude. a** Experimental NMOR signals versus the amplitude of a tilted magnetic field. The first harmonic signal and second harmonic signal is shown in a1 and a2 respectively. The laser is frequency modulated at 997 Hz, with a modulation range of 400 MHz. The center frequency of the laser is 200 MHz red-detuned from $^{87}$Rb D$_1$ line, $F = 2 \rightarrow F' = 1$ transition. The laser power is 20 μW. X is the in-phase signal and Y is the quadrature signal. **b** Theoretically calculated NMOR signals as a function of magnetic field. The first harmonic signal and second harmonic signal is shown in b1 and b2 respectively. In all figures, the red (blue) curve corresponds to the X(Y) signals and left (right) $y$-axis.

signals exhibit resonance at $\Omega_L = 0$, $\omega_m$ and $\frac{1}{2}\omega_m$. The resonance center can be found precisely by fitting the curves with a generalized Lorentzian function, which is the key in coil calibration. For vector magnetometry, we choose the resonance at $\Omega_L = \omega_m$, since more EIT channels take part in the interferences than the $\Omega_L = \frac{1}{2}\omega_m$ resonance, as can be seen from Fig. 1a, allowing more information to be encoded. Figure 2b shows the spectrum calculated by the 8-level theoretical model using the master equation. Despite of the qualitative agreement, the experimental spectra deviate from the theory results because it is impractical to include in the model the accurate information of the following experimental complications which affect both the resonance lineshape and the absolute signal values: (a) demodulation phases are unknown in the phase sensitive detection due to phase delays in the electronics. (b) the input light polarization is slightly altered by the cell window. (c) there is a wide pedestal for the narrow NMOR resonance, charateristic of the coated cell and related to the thermal motion of the atoms[55–58]. We emphasize that due to motional averaging[57], the field inhomogeneities of the coil causes negligible line broadening, as evidenced in our experiment by the zero-power resonance linewidth[55] of 1 Hz for both the resonances at $\Omega_L = \omega_m$ and $\Omega_L = \frac{1}{2}\omega_m$, which is likely dominated by spin exchange. Because of

the above reasons, relying on the master equation theory model in establishing the relation between the signals and the **B** field parameters is generally not suitable, while the NN can provide a better solution.

Then we train the the NN using NMOR signals for a large range of field amplitudes and orientations. The structure of the fully connected NN is shown in Fig. 3a. There is one input layer receiving the four-dimensional NMOR signal and one output layer releasing the field information. Between the input and output layer there are 8 hidden layers each containing 128 neurons and the L2 regularization[59] is used to prevent over-fitting. The activation function in the hidden layer is a ReLU (rectified linear unit) function[60]. The data set is divided into the training set and verification set in the proportion of 8 to 2 and the mean squared error is defined as the loss. The training set is used for learning, i.e., to determine the weights in the NN, while the validation set is used to assess the performance of the already trained NN. In practice, the NMOR data at the input layer for training is generated by a reverse-NN[11] with a similar structure. After using the $(B, \theta, \varphi)$ set as the input and the corresponding experimental data $(X_1, Y_1, X_2, Y_2)$ as the output for training, this reverse-NN can be employed to produce optical rotation data which is denser and more robust against noise

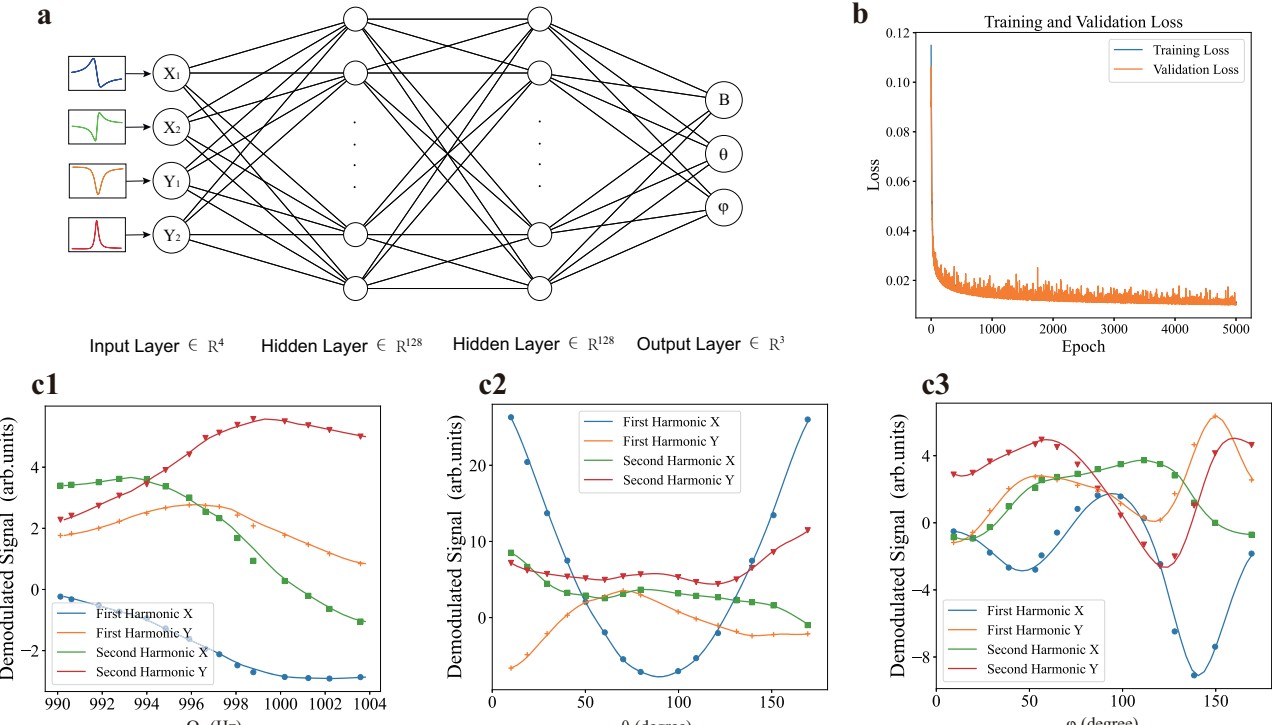

**Fig. 3 | Architecture and performance of the neural network (NN). a** Illustration of the neural network. The demodulated optical rotation signals' quadratures X and Y at the first and second harmonics of $\omega_m$ form the 4-dimensional input. The NN gives the magnitude of the magnetic field $B$ and its direction $\theta, \varphi$ as output. **b** Training process of the NN. Loss of the training set and validation set decreases with the rounds of iteration. Mean squared error is used as the loss function. There

is no obvious difference between the training loss and validation loss which means no over-fitting. **c** Test of the validity of NN. Scattered points are predictions from the trained NN and solid lines are the dense reproduction of the input data through an inverse NN (see *text*), which show good agreement. In c1: $\theta = 60°$, $\varphi = 60°$, in c2: $\varphi = 60°$, $\Omega_L = 997$ Hz, and in c3: $\theta = 60°$, $\Omega_L = 997$ Hz.

than the measured. We then use these denser NMOR data to train the NN as shown in Fig. 3a with an Adam optimizer[61], and the training and validation error is plotted in Fig. 3b. The trained NN can reproduce the full vectorial information of the magnetic field accurately as shown in Fig. 3c, where the solid lines are data generated from the reverse NN and the scattered points are from the prediction of the NN. In our data set, we have chosen the range for $\theta$ and $\varphi$ to be [10°,170°], because the NMOR signals are insensitive to the variation of $\varphi$ ("dead zone") when **B** is nearly aligned with the propagation direction of the light **k** ($\theta \approx 0°$ and 180°). One other issue is the signal degeneracy for $\varphi$ and $\varphi + \pi$, but we propose an angled multi-pass configuration to lift this degeneracy and also to remove the "dead zone" for $\varphi$ (see Supplementary Note 6).

Finally, we examine the sensitivities of the three polar components $B, \theta, \varphi$ given by our NN scheme. The normal way to obtain the magnetometer sensitivity is to convert the fluctuations on the measured signal $\delta S$ to that on the magnetic field $\delta B$ through a measured slope $dS/dB$. Here, an analogous "slope" is provided by the trained NN which establishes a map between the optical rotation signals and magnetic field parameters. We continuously record the signal set of optical rotations $(X_1, Y_1, X_2, Y_2)$ for about one minute at a sampling rate of 900 per second for each fixed **B**, and the signal set at each time point is fed to the NN which then outputs the predicted parameter $(B, \theta, \varphi)$. Consequently, the four time traces of the signals $X_1(t), Y_1(t), X_2(t), Y_2(t)$ are converted into three time traces $B(t), \theta(t), \varphi(t)$. We then perform fast-Fourier-transform (FFT) on $B(t), \theta(t), \varphi(t)$ respectively, and obtain the sensitivities, where the frequency response has also been considered and was obtained experimentally with the aid of the NN (see Supplementary Note 4) using a similar approach as described here.

Shown in Fig. 4a are the sensitivities at low frequencies for an exemplary **B** field direction of $\theta = 63.435°$, $\varphi = 60°$ with an amplitude of

about 140 nT, while we found that in other field orientations the sensitivity is at a similar scale (see Supplementary Note 5). Due to the relatively small bandwidth of our magnetometer (associated with the narrow linewidth ~16 Hz of NMOR resonance), sensitivities are better at lower frequency. The best sensitivities are observed in the range of 10–20 Hz, where the sensitivity of field magnitude is about 100 fT/$\sqrt{\text{Hz}}$, and the angular sensitivity has the order of 100 $\mu rad/\sqrt{\text{Hz}}$. The extra noise at low-frequency near DC is mainly from the magnetic field itself, as well as $1/f$ noises. In order to confirm the sensitivities given by the NN, we examined whether a small change at these sensitivity levels in the magnetic field can be detected. We applied a small AC magnetic field at 11 Hz to slightly vary $(B, \theta, \varphi)$, and the NN is trained for the AC field in the parameter space near $B \approx 140$ nT ($\Omega_L \sim 997$ Hz), $\theta = 63.435°$, $\varphi = 60°$. The test field change has an interval of (140 fT, 0.02°, 0.02°). The predicted changes in the vector components of **B** are consistent with the true values, as shown in Fig. 4b where the sizes of the error bars (standard deviations) indicate the sensitivities, which agree with those given by the NN-aided noise analysis shown in Fig. 4a. These results prove that ML-assisted approach for vector magnetometry can give the correct sensitivity levels.

## Discussion

We propose a paradigm for atomic vector magnetometry based on machine learning, allowing three dimensional single-shot information extraction using a simple standard scalar magnetometer setup. Acquiring the amplitude and phase of the AC optical rotation signals removes the need for spectral sweep, enabling future real-time measurement of time varying magnetic field. The single-beam all-optical design is suitable for dense integration of the sensor units. We also demonstrate how to obtain vector field sensitivities using the neural network, and the best sensitivities on field amplitude and orientations

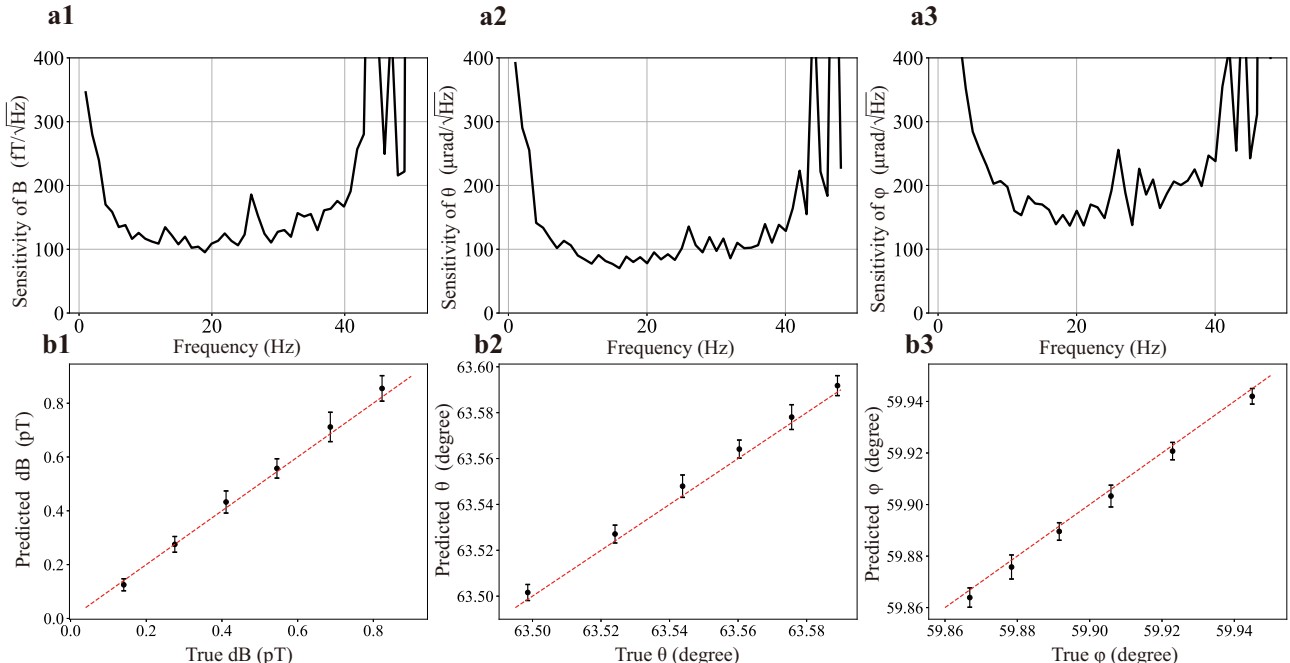

**Fig. 4 | Sensitivity of the machine learning assisted vector magnetometer.**
**a** Neural network predicted sensitivity for field amplitude (a1) and orientations (a2, a3) at low frequency. The measurement is performed at $\theta = 63.435°$, $\varphi = 60°$ for a field magnitude about 140 nT. **b** NN-predicted change of the magnetic field magnitude (on top of 140 nT, b1) and directions (b2, b3) versus the corresponding true values. The dashed line corresponds to the $y = x$ function. The results are demonstrated for magnetic field changes at a frequency of 11 Hz. The size of the error bars (standard deviation from 60 repetitive independent measurements) are in agreement with the NN predicted sensitivity level at 11 Hz.

are about $100\ \text{fT}/\sqrt{\text{Hz}}$ and $100 \sim 200\ \mu rad/\sqrt{\text{Hz}}$ respectively. The current sensitivities are limited by electronic noises around the relatively low modulation frequency. After removal of such noises, the sensitivity may be further improved using a multipass design[62]. The signal degeneracy for $\varphi$ and $\varphi + \pi$ can be lifted with an angled multipass configuration, as shown in our simulation (see Supplementary Note 6), which also removes the dead zone for $\varphi$ when **B** is nearly aligned with **k** of the laser. Furthermore, the dynamic range of detectable magnetic field can be controlled through the resonance linewidth or changing the modulation frequency of the laser. Higher bandwidth can be obtained in vapor cells working in the higher temperature spin-exchange-relaxation-free regime[28].

Our strategy of using machine learning to simplify the structure of vector NMOR-magnetometers can be extended to other types of atomic magnetometers, as well as multiparameter sensors in general, using the following procedure: (1) Identify a set of observables which are sensitive to the target parameters and can be simultaneously, if possible, recorded in the experiment. The rich degrees of freedom in the interrogating laser or broadly the electromagnetic field, for example the amplitude, polarization, spatial modes, frequency spectra etc., can be all used for encoding the information indirectly and compressively. (2) Stabilize the experiment system as a prerequisite for a robust map between the observable set and the parameter set. (3) Experimentally collect data within a suitable range of target parameters and perform the neural network training to build the map between the signal set and parameter set. The NN structure is chosen according to the complexity level of the problem, and overfitting should be avoided. (4) Conduct real measurements using the trained NN.

## Methods
### Theoretical model
Our numerical calculation used the eight-level atomic system as shown in Fig. 1 in the main text. However, since our simulations showed that

the four-level model gave qualitatively similar results as the eight-level model, to gain intuition on the key physics, we here describe a simplified four-level system, as shown in Fig. S1, where the ground states have three Zeeman levels which couple to one excited state by $\sigma^+, \pi, \sigma^-$ polarized light fields respectively. The atom-light interaction Hamiltonian $H$ can be derived with the rotating wave approximation (RWA), and the atomic coherences can be found from the density matrix $\rho$ by solving the master equation:

$$\frac{\partial \rho}{\partial t} = -\frac{i}{\hbar}[H,\rho] + \left(\Gamma_{\text{rel}} + \Gamma_{\text{rep}}\right)\rho, \tag{1}$$

where $\Gamma_{\text{rel}}$ describes the decoherences including the spontaneous decay and dephasing etc., and $\Gamma_{\text{rep}}$ describes the repopulation of the ground states[63]. Due to the periodicity of the system under frequency modulation, the coefficients of a Fourier expansion of the density matrix can be identified using the Floquet technique where $\rho(t)$ is expanded in harmonics of the modulation frequency $\omega_{\text{m}}$:

$$\rho(t) = \sum_{n=-\infty}^{\infty} \rho^{(n)} e^{in\omega_{\text{m}}t} \tag{2}$$

Then the polarization rotation signal of the light we measure can be derived from the atomic coherences, which is found to contain the full vectorial information of the magnetic field. More details are in the Supplementary Note 1.

### Calibration of magnetic field
In the experiment, the magnetic field to-be-measured is provided mainly by the three sets of orthogonal Helmholtz coils within the shields, where precise calibration is required in order to generate a magnetic field along any direction as we intend. In the calibration process, we obtain the amplitude of the total magnetic field (produced by the coils and background magnetic field in the shields) by

identifying the resonance locations of the NMOR spectrum obtained through slowly sweeping the laser modulation frequency $\omega_m$. As shown in Fig. S2, the spectra exhibit resonance when the Larmor frequency $\Omega_L$ equals $\omega_m$ (or $\frac{1}{2}\omega_m$, not shown). The resonance center is found by fitting the experiment curve with a linear superposition of a Lorentzian absorption and dispersion function. For a single set of Helmholtz coil, the relation between the current applied and the generated magnetic field is linear. However, for the vector synthesis of a magnetic field generated by three sets of coils, imperfection in the orthogonality of the coils should be considered. Furthermore, the residual background magnetic field in the magnetic shields couldn't be neglected.

The strategy we used for calibration is similar to that used in reference[64]. We consider a coil system with imperfect orthogonality among the three sets of coils which yield magnetic fields $B_{X_c}, B_{Y_c}, B_{Z_c}$ along $\mathbf{X}_c, \mathbf{Y}_c, \mathbf{Z}_c$ axis respectively, as shown in Fig. S3. First, for each set of coil we obtain the relation between the field amplitude and the current through the NMOR spectra with only this coil in operation. Then, without losing generality, we can set small angles $\xi, \eta, \zeta$ (see Fig. S3) to describe the deviation of $(X_c, Y_c, Z_c)$ from a normal orthogonal coordinate system $(X, Y, Z)$, and we have:

$$\mathbf{X}_c = \begin{pmatrix} \cos\xi \\ 0 \\ \sin\xi \end{pmatrix}, \mathbf{Y}_c = \begin{pmatrix} \sin\eta\cos\zeta \\ \cos\eta\cos\zeta \\ \sin\zeta \end{pmatrix}, \mathbf{Z}_c = \begin{pmatrix} 0 \\ 0 \\ 1 \end{pmatrix}. \quad (3)$$

The total magnetic field is $\mathbf{B} = B_{X_c}\mathbf{X}_c + B_{Y_c}\mathbf{Y}_c + B_{Z_c}\mathbf{Z}_c + \mathbf{B}_{\text{residual}}$, which can be written as:

$$\begin{aligned} B_{X_c}\cos\xi + B_{Y_c}\sin\eta\cos\zeta + B_{X_0} &= B\sin\theta\cos\varphi \\ B_{Y_c}\cos\eta\cos\zeta + B_{Y_0} &= B\sin\theta\sin\varphi \\ B_{X_c}\sin\xi + B_{Y_c}\sin\zeta + B_{Z_c} + B_{Z_0} &= B\cos\theta \end{aligned} \quad (4)$$

or:

$$\begin{aligned} &(B_{X_c}\cos\xi + B_{Y_c}\sin\eta\cos\zeta + B_{X_0})^2 \\ &+ (B_{Y_c}\cos\eta\cos\zeta + B_{Y_0})^2 \\ &+ (B_{X_c}\sin\xi + B_{Y_c}\sin\zeta + B_{Z_c} + B_{Z_0})^2 = B^2. \end{aligned} \quad (5)$$

Here $B, \theta, \varphi$ are respectively the amplitude, altitude angle and azimuth angle of the total magnetic field we intend to measure. $B_{X_0}, B_{Y_0}, B_{Z_0}$ are the components of the residual magnetic field along $\mathbf{X}, \mathbf{Y}, \mathbf{Z}$ respectively. The total magnetic field's amplitude $B$ as expressed by Eq. (5) can be measured from the NMOR spectra. By traversing the currents in the three coils and measuring the total field amplitude $B$ for each set of $(B_{X_c}, B_{Y_c}, B_{Z_c})$, we can determine parameters $(\xi, \eta, \zeta, B_{X_0}, B_{Y_0}, B_{Z_0})$ using Eq. (5) through non-linear least squares fitting. Then, to set a total magnetic field with parameters $B, \theta, \varphi$ as we intend, we can solve Eq. (4) to find what magnetic field should be generated in each coil, i.e., $(B_{X_c}, B_{Y_c}, B_{Z_c})$.

### Implementation of neural network
Neural Network (NN) is an artificial intelligence (AI) method based on the connectivism which imitates the connection between neurons. Our model is a simple fully connected Neural Network, and we proceed as follows to mimic the function of the biological neural network. First, data are collected in pairs of feature (input) and label (output). Commonly, the larger the amount of data, the better the performance of the NN. Second, we build the structure of the NN with a complexity determined by the scale of the problem to be solved. Similar to the growth of cognitive ability of human, the NN receives large amount of collected data with features and corresponding labels which change the weights of neurons. The NN updates its parameters via backpropagation using gradient descent algorithm aimed to reduce the

loss function we choose. This is the training process of the NN. In our experiment mean-squared error is chosen to be the loss function. After training, parameters in the NN are fixed and new data of features can be sent to the input port of the NN and it will output the predictions.

Our Neural Network is implemented using the framework of Keras, a high-level API (Application Programming Interface) of Tensorflow written in python. In Keras, a model is understood as a sequence or diagram composed of independent and fully configurable modules. These modules can be assembled together with as few restrictions as possible. In particular, modules such as Neural Network layer, loss function, optimizer, initialization method, activation function, and regularization method, can be combined to build new models.

The input layer of our NN receives the four-dimensional NMOR signal $(X_1, X_2, Y_1, Y_2)$ and the NN predicts the three-dimensional magnetic field information $(B, \theta, \varphi)$ as the output. Between them are 8 hidden layers each containing 128 neurons. The transmission between layers is implemented via matrix operation and in each neuron there should be a non-linear activation function. ReLU activation function is used in each neuron. Mean-squared error is chosen to be the loss function, and additional term is added to the loss function to prevent overfitting. By calling the Keras API for L2 regularization in the hidden layers, quadratic sum of all the parameters in the hidden layers are recorded and added to the loss function. This procedure guarantees the generalization ability of the model, i.e., it will prevent overfitting which often means a complicated NN that adjusts the input and output relation only for the training data set. As for the training process, the adaptive moment estimation method[65] is applied.

### Data availability
The data supporting the findings of this study are included in the paper and its Supplementary Information. The NN training data used in this study are available in Github(https://github.com/XinMeng95/Machine-Learning-Assisted-Vector-Atomic-Magnetometry/).

### Code availability
The NN code for this study is available in Github (https://github.com/XinMeng95/Machine-Learning-Assisted-Vector-Atomic-Magnetometry/).

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

## Acknowledgements

We thank Lei Shi, Tongyu Li, Lingjie Fan, Guiying Zhang, Kai-Feng Zhao, Yue Ban and Klaus Mølmer for helpful discussions. This work was supported by NNSFC under Grant No. 12027806 (Y.X.), Shanxi "1331 Project" (Y.X.) and Packard Foundation 2020-71479 (L.J.).

## Author contributions

Y.X. conceived the idea and supervised the work. X.M., Y.Z., X.Z., Sh.J. and T.W. constructed the experiment. X.M. and Y.Z. carried out the experiment, numerical simulation and data analysis under Y.X.'s direction. X.M., Y.Z. built the machine learning model with L.J.'s guidance. X.M., X.Z. and Y.X. wrote the manuscript. Su.J., L.X. and all other authors discussed the experiment design, results and contributed to the manuscript.

## Competing interests

The authors declare no competing interests.
