## [Peer Review File · Nature Communications]

Reviewers' comments:

Reviewer #1 (Remarks to the Author):

I have read the manuscript entitled "Machine Learning Assisted Vector Atomic Magnetometry", submitted to Nature Communications with the code NCOMMS-23-00060.

The work describes an optical magnetometer in which the output signal is analyzed by means of neural network methodology to obtain a vectorial response thanks to a machine learning approach.

The subject is definitely attractive and innovative for its inherent multidisciplinary content, however I have severe doubts about the general interest of the work to a broad community and/or the particular interest of a narrower audience, which would not find sufficiently deep details to improve their knowledge. I also noticed some important weaknesses, that should be highlighted more clearly and that probably make the results not as promising as expressed in the introduction.

For these reasons I do not consider this manuscript worthy of being considered for publication in a general journal such as Nat. Comm., and, at the same time I encourage the Authors to publish their results in a more specialized journal, after having completed it with important information that is presently missing.

My point is that the reader is here informed about interesting achievements obtained by the convergence of mature disciplines, but he will not gain additional know-how with the necessary details to reproduce or to continue this research. Actually the OPM apparatus consists in a standard setup, and the ML approach is implemented with standard techniques: the message is that this convergence is successful, but the reader cannot find sufficient information to reproduce the experiment or to learn original approaches/ concepts to be applied in his own (similar) apparatus.

In any case, I wish to list some remarks aimed to improve the readability. These remarks are in addition to the more profound criticism expressed above, which indeed requires an important expansion with details about the ML and NN implementation.

Page 1

Which may not be suitable for real-time field measurement

strictly speaking, real time does not mean "nearly instantaneous" but "responding with an assigned delay": in this sense, the single-shot (thus very fast) response is precluded, while a real-time one could be possible.

Page 3

principle

Summarizing, spherical co-ordinates are used, with the polar axis oriented along the k direction. There is a missing information about the azimuthal angle origin: how is the $\phi=0$ defined?

experiment setup

(I would expect "experimental setup") is really the observed linewidth as narrow as 0.5Hz? What is the theoretically expected value, taking into consideration residual field inhomogeneities and the various relaxation processes? Isn't this half Hz linewidth in relation with the 16Hz reported in page 5, 2nd column?

Page 4

caption of fig.2

a) range->depth (?)

b) 200MHz red detuned: how is this value selected/determined? Why?

c) X is the in-phase quadrature and Y is the out-of-phase quadrature : with the usual terminology this should be probably reworded as:

"X is the in-phase signal and Y is the quadrature signal."

d) "...curve corresponds to the red (blue) y-axis":

this should probably be reworded as "to the X (Y) = in phase (in quadrature) signals"

Therefore, relying on the master equation theory model in establishing the relation between the signals and the B field parameters is generally not suitable, while the NN can provide a better solution.

This sentence clarifies and puts in evidence a crucial weakness of the proposed methodology. If I understand correctly, this means that the ML approach enables measurements despite the field imperfections, because the system is trained with a set of magnetic field values having the same inhomogeneities that will characterize the fields to be measured. If so, such a device could only measure fields applied (inside the shield) by the same coils used to generate at the training stage. In this sense, it will not be a magnetometer, but an amp-meter measuring the current flowing in those coils.

relu (ReLU): the acronym should be defined

Page 5

the NMOR signals are insensitive to the variation of phi, making the NN difficult to converge.

this issue is real, it is set by the physical nature of the system: the "difficulty to converge" is actually due to information that is not contained in the acquired data. This implies that, under some geometries (B and k nearly aligned), the actual instrumental resolution is much worse than what is declared (e.g. in the 2nd column), and this is not dependent on the approach used to analyze the magnetometric signal.

The paragraph in the first half of the second column is unclear or very hard to understand. It likely requires a complete reformulation.

Page 6

We applied a small AC magnetic field at 11 Hz to slightly vary (B, theta, phi)

11Hz is not \ll the 16Hz bandwidth mentioned in page 5, thus one expects that the system response is no longer that measured under DC condition at the training stage: either the training should be made also under dynamic conditions, or the frequency range in which the DC training is assumed to be valid should be defined better.

DISCUSSIONS -> *DISCUSSION*

This section seems more a "conclusion" than a "discussion".

Quantitative assessments would be expected in a discussion: as an example, the sentence "...the neural network, and achieve good sensitivities on field amplitude..." should be made quantitative, replacing "good" with a number.

The bandwidth of the magnetometer can be increased by broadening the resonance with higher laser power.

One would expect that after increasing the signal by means of cell heating, the spin exchange would make the resonance broader with no need of using power broadening. And noticeably this will rapidly lead to increase the signal amplitude, but not to enhance the sensitivity. As a general remark, such "technical" improvements should be made before collecting data for publication: I strongly suggest to apply NN and ML to already optimized magnetometric signals.

In ref.7, the name Munoz requires a tilde.

Reviewer #2 (Remarks to the Author):

The manuscript reports on an atomic vector magnetometer in a single optical beam geometry that uses an artificial neural network (NN) to map four simultaneous NMOR resonance signals, namely the in-phase and out-of-phase quadrature components at the modulated frequency and second-harmonic, to the magnetic field strength and direction with 100 fT/rt[Hz] and 100 uRad/rt[Hz] sensitivities. The authors demonstrate these results with a 140 nT field generated by a calibrated 3D coil system. This work is noteworthy because the vector information is acquired simultaneously in a single shot with a single beam geometry, which simplifies the sensor architecture compared to current vector magnetometers.

The quality of the data presented here is excellent and support the reported results. The paper is

well-written, except for minor comments below, and previous work is properly cited. Together with the supplementary material, this manuscript gives all the necessary experimental and theoretical details. The vector sensitivities reported are near the state of the art but are not notable by themselves. In addition, the NMOR techniques are also standard and the NMOR signals used in this work have been historically reported by Budker's group to be useful for vector magnetometry. The main innovative step in this work is to apply a NN to properly map these NMOR signals to the magnetic field vector parameter space.

By using ML to effectively simplify the sensor's architecture, this work will surely impact the field of atomic magnetometry. For vector magnetometers a simpler architecture means the sensor is less prone to drifts and can be more compactly designed. This work may also be of interest to other disciplines outside of magnetometry that require multi-parameter sensing.

In view of the above, I recommend this paper for publication with some minor comments and suggestions below:

- I suggest that the font size in the figures should be increased for better readability. This is especially true for the plot axes in Figures 2,3,4
- In Sec. II, second paragraph "...weights updated using [the] gradient descent algorithm..."
- The "relu" function in Sec. IV paragraph three is often written as "ReLU".
- Often there is a missing "the" before "NN". E.g. in last paragraph of sec. IV "... is in good agreement with those given by [the] NN method".
- In Fig. 4 b1, I suggest writing something like dB instead of just B since it is a change of the magnetic field value that you are plotting here.
- Revise grammar and wording in the sentence "The predicted vector...method" of the last paragraph of section IV.

Dear editor,

Thank you for your great patience in handling our manuscript "Machine Learning Assisted Vector Atomic Magnetometry" with code NCOMMS-23-00060.

We thank both referees for carefully reading the manuscript and providing critical and helpful comments. Below please find our point-to-point response (in blue) to all the questions and suggestions.

Sincerely,
The authors.

Referee#1

I have read the manuscript entitled "Machine Learning Assisted Vector Atomic Magnetometry", submitted to Nature Communications with the code NCOMMS-23-00060.

The work describes an optical magnetometer in which the output signal is analyzed by means of neural network methodology to obtain a vectorial response thanks to a machine learning approach.

The subject is definitely attractive and innovative for its inherent multidisciplinary content, however I have severe doubts about the general interest of the work to a broad community and/or the particular interest of a narrower audience, which would not find sufficiently deep details to improve their knowledge. I also noticed some important weaknesses, that should be highlighted more clearly and that probably make the results not as promising as expressed in the introduction.

Response:

We thank the referee for recognizing the “innovative” feature and the “inherent multidisciplinary content” of the subject of this study.

The general interest of this work can be summarized as the following, quoting referee #2's report : “By using ML to effectively simplify the sensor's architecture, this work will surely impact the field of atomic magnetometry. For vector magnetometers a simpler architecture means the sensor is less prone to drifts and can be more compactly designed. This work may also be of interest to other disciplines outside of magnetometry that require multi-parameter sensing.”

As suggested by the referee, to help the audience “improve their knowledge” and apply our method in their experiments, in the revised manuscript we have supplied more details. In particular, a new section on ML is added in the supplementary material, and a “recipe-like” description of how to apply our methodology in other sensors is added in the last paragraph of

the main text. Various other details about the experiment have also been added in the revised main text.

We have also responded to the two weak aspects pointed out by the referee, (a) Regarding the insensitivity to ϕ when B is nearly along k: in the revised manuscript, we have proposed ways to eliminate this “dead zone” problem. (b) Regarding that the trained NN may not work for fields produced by sources different than our test coils: we explain that this is a misunderstanding, and gave a list of factors contributing to the difference in the master equation simulation and the experiment, using new experiment data, simulation results and references as evidence. Please see the details in our point-to-point response below.

For these reasons I do not consider this manuscript worthy of being considered for publication in a general journal such as Nat. Comm., and, at the same time I encourage the Authors to publish their results in a more specialized journal, after having completed it with important information that is presently missing.

My point is that the reader is here informed about interesting achievements obtained by the convergence of mature disciplines, but he will not gain additional know-how with the necessary details to reproduce or to continue this research. Actually the OPM apparatus consists in a standard setup, and the ML approach is implemented with standard techniques: the message is that this convergence is successful, but the reader cannot find sufficient information to reproduce the experiment or to learn original approaches/ concepts to be applied in his own (similar) apparatus.

Response:

We thank the referee for carefully reading our manuscript, and for pointing out issues in the presentation, such as lack of various details. In the revised version, we have added more information as suggested.

Our experiment apparatus is a standard one used in a NMOR magnetometer, and we have given all the experiment details including the experimental parameters such as laser power, detuning, modulation parameters, laser polarization and detection techniques. With the information in our manuscript, we believe that our experiment can be reproduced by an interested reader. Regarding the details of the ML, we have added more information in the supplementary material. As for how to apply our approach in other multiparameter sensing experiments, we have added a “recipe-like” description in the last paragraph of the main text. Moreover, we are happy to share our code upon request.

As pointed out by the referee#2 in the report, “Together with the supplementary material, this manuscript gives all the necessary experimental and theoretical details.” Now, with the additionally provided details in the revised manuscript, we believe that a general reader can learn how to use our proposed methodology in his/her own experiment design.

In any case, I wish to list some remarks aimed to improve the readability. These remarks are in addition to the more profound criticism expressed above, which indeed requires an important expansion with details about the ML and NN implementation.

Response: In the revised manuscript, we have added a section in the supplementary material with details of our ML and NN implementations

Page 1

Which may not be suitable for real-time field measurement

strictly speaking, real time does not mean "nearly instantaneous" but "responding with an assigned delay": in this sense, the single-shot (thus very fast) response is precluded, while a real-time one could be possible.

Response: We thank the referee for the comment. It is true that "single-shot" may indicate a very fast response, and indeed any atomic sensor's response is limited by the dynamics of the atom-light interaction. In the schemes requiring a sweep of the spectrum, the to-be-measured field needs to be constant during the sweep time, which is approximately equal to the optical pumping time (the time it takes for the atomic system to reach steady state) MULTIPLIED BY the number of points in the sweep. In our case, the field only needs to be nearly constant within the optical pumping time.

As a response to this concern from the referee, we have rephrased the relevant sentences. We changed the sentence "Which may not be suitable for real-time field measurement" to "Which may not be suitable for relatively fast or real-time field measurement". Also, when the word "single-shot" first appears in the manuscript, we added a note "within the sensor's response time".

Page 3

principle

Summarizing, spherical co-ordinates are used, with the polar axis oriented along the k direction. There is a missing information about the azimuthal angle origin: how is the $\phi=0$ defined?

Response: We thank the referee for pointing this out. We have added the definition of $\phi = 0$ in our revised manuscript. It is defined by the test coils along the x direction, which is also the polarization-beam-splitter's horizontal axis.

experiment setup

(I would expect "experimental setup") is really the observed linewidth as narrow as 0.5Hz? What is the theoretically expected value, taking into consideration residual field inhomogeneities and the various relaxation processes? Isn't this half Hz linewidth in relation with the 16Hz reported in page 5, 2nd column?

Response:

The linewidth of the NMOR resonance is the sum of two parts: one part is the intrinsic linewidth determined by the ground state coherence decay, such as dephasing due to the magnetic field

inhomogeneity, atom-wall collisions and atom-atom collisions; and the other part is the power-broadened width, proportional to the laser power. Here, 0.5Hz is the zero-laser-power EIT linewidth (*this is half-width at half maximum, we now change it to full-width at half maximum, 1Hz, for consistency with the 16Hz power broadened full-width*), extracted from a linewidth versus laser-power measurement shown in the figure below.

Fig.R1 Zero-power linewidth (full width at half maximum) extraction of the NMOR resonance. Left: resonance near zero-magnetic-field. Right: resonance near 70nT (blue lower curve) and 140nT (red upper curve). These resonances have different width for the same (nonzero) laser power because different polarization components participate the resonances.

It can be seen that, the 70nT and 140nT resonances have nearly the same zero-power resonance of about 1Hz (full width at half maximum), which indicates that the 1Hz is not dominated by field inhomogeneity from the coils (otherwise the 140nT resonance's zero-power width would be about twice that of the 70nT resonance), but dominated by spin exchange at room temperature [In literature, Phys. Rev. Lett. 81, 5788 (1998), a similar measurement was made for ^{85}Rb]. We note that, the zero-field resonance has a larger zero-power linewidth, because it is dominated by field inhomogeneity from the ambient field whose effect is greatly reduced when there is a much larger field (for instance for the 70nT and 140nT case). It has been established in literature that, the existence of a relatively large field can suppress the dephasing effect due to ambient field inhomogeneities [for example, see Phys. Rev. A 72, 023401 (2005); Phys. Rev. A 79, 043815 (2009)].

We emphasize that, here motional averaging in wall-coated cell has played an important role in mitigating the line broadening effect due to magnetic field inhomogeneities of the ambient and the coils. Using a commercial fluxgate magnetometer, we measured the ambient field inhomogeneity inside the shields within the vapor cell volume, and found it to be about 1nT. Moreover, our test coils are carefully designed and the fractional inhomogeneities within the cell is calculated to be about 8/1000, which means that for a 140nT field, the inhomogeneity is about 1nT. According to the g factor of ^{87}Rb , field inhomogeneity of 1nT within the cell should correspond to 14 Hz of broadening (full width) in our NMOR zero-power width if there is no motional averaging, but experimentally we extracted a zero-laser-power FWHM (full width at half maximum) of 1Hz (non-zero-field resonance) and 2Hz (zero-field resonance), indicating the effect of motional averaging. The quality of the alkene coating determines the extent of

motional averaging of the inhomogeneous magnetic field. The coating's quality varies from cell to cell because of the somewhat uncontrollable chemical process, so the associated line broadening can only be measured and there is no theoretical value available. The 16Hz reported in page 5, 2nd column is the FWHM (full width at half maximum) of the EIT resonance at 20 μ W input laser power, i.e., this is a power-broadened linewidth.

In the revised manuscript, we have added a sentence about power broadened width of 16Hz where the 0.5Hz (now 1Hz full-width at half maximum) width was first mentioned. We also added some information about the ambient field, coil field and motional averaging.

Page 4

caption of fig.2

a) range->depth (?)

Response: Here 400MHz is not the modulation depth but twice the frequency modulation amplitude, i.e., 400MHz is the range through which the laser frequency is swept during the frequency modulation. In the revised manuscript, we added a note "or, modulation amplitude of 200MHz".

b) 200MHz red detuned: how is this value selected/determined? Why?

Response: The center frequency of the laser is optimized for the NMOR signal. It is red detuned to partially avoid the influence the $^{87}\text{Rb } D1 |F = 2\rangle \rightarrow |F' = 2\rangle$ transition, whose dark state (a linear superposition of the ground states) is orthogonal to that of, or is the bright state for, the $^{87}\text{Rb } D1 |F = 2\rangle \rightarrow |F' = 1\rangle$ transition due to different signs in the C-G coefficients. In the revised manuscript, we have added two references [Phys. Rev. A 73, 053404 (2006); Appl. Phys. Lett. 119, 054001 (2021)] where such optimization of laser detuning has been applied too. The value of the optimized detuning depends on laser power, Doppler broadening etc.

c) X is the in-phase quadrature and Y is the out-of-phase quadrature: with the usual terminology this should be probably reworded as:

"X is the in-phase signal and Y is the quadrature signal."

Response: We thank the referee for pointing this out. We have adopted the suggested terminology in the revised manuscript.

d) "...curve corresponds to the red (blue) y-axis":

this should probably be reworded as "to the X (Y) = in phase (in quadrature) signals"

Response: We thank the referee for pointing this out. We have made the change as suggested.

Therefore, relying on the master equation theory model in establishing the relation between the signals and the B field parameters is generally not suitable, while the NN can provide a better solution.

This sentence clarifies and puts in evidence a crucial weakness of the proposed methodology. If I understand correctly, this means that the ML approach enables measurements despite the field imperfections, because the system is trained with a set of magnetic field values having the same inhomogeneities that will characterize the fields to be measured. If so, such a device could only measure fields applied (inside the shield) by the same coils used to generate at the training stage. In this sense, it will not be a magnetometer, but an amp-meter measuring the current flowing in those coils.

Response: We understand this concern, but we think there is a misunderstanding here, mainly due to the lack of details in our previous manuscript. Below we give an in-depth description of the major factors accounting for the discrepancy between the master equation simulation and the experiment. These factors have also been listed in the revised manuscript.

First of all, we emphasize that, due to motional averaging in our wall-coated vapor cell, the inhomogeneities in the field from both the coil and the ambient cause negligible line broadening, in particular, less than 1Hz for the resonance near 140 nT which we use for the magnetometer demonstration. As evidence, we obtained the same zero-power width of 1Hz for the 70nT and 140nT resonance (Fig.R1 above), which indicates that field inhomogeneities from the coils are not a concern for the magnetometer demonstrated here. Detailed analysis and the figures on the zero-power linewidth is in our response (surrounding Fig.R1) to the referee comment on this regard.

Secondly, we point out that it is the following three factors that mainly account for the discrepancy between the computed and the measured NMOR spectra.

- (1) Demodulation phases in the phase sensitive detection. In our experiment, there is phase lag between the signal driving the laser's frequency modulation (through the laser's PZT, i.e., Piezoelectric Transducer) and the output optical rotation signals' oscillation; also, our lock-in amplifier uses the PZT's driving signal as the external reference and the default demodulation phase is set to be zero. Therefore, the theoretical demodulation phase is unknown, and is adjusted in the model to best match the measured lineshapes. Because the lineshape is also affected by other factors (see below (2) and (3)), the demodulation phase cannot be accurately identified. We show below both experimentally and theoretically that a change in the demodulation phase can alter the lineshape of the in-phase and quadrature signals

Fig.R2 Experimental comparison between resonance lineshapes at two demodulation phases with 60° difference.

Fig.R3 Theoretical comparison between resonance lineshapes at two demodulation phases with 60° difference.

(2) Polarization imperfection of the input light. Since the NOMR lineshape is the result of the interference of many Lambda-type EIT as shown in Fig.1 of the main text, both the phases and amplitudes of the σ^+ , σ^- and π polarization components matter. The front glass window of the vapor cell can cause a small unknown polarization change to the input light, so using the polarization state measured before the cell in the simulation can lead to discrepancy with the experiment result. We show both experimentally and theoretically that a small change in light polarization does lead to lineshape change.

Fig.R4 Experimental comparison between resonance lineshapes when the input laser's polarization direction is altered by 3 degrees (a rotation of 6 degrees about the z-axis on the Poincare sphere).

Fig.R5 Theoretical comparison between resonance lineshapes when the input laser's polarization direction is altered by 3 degrees (a rotation of 6 degrees about the z-axis on the Poincare sphere).

Fig.R6 Modified simulation results compared to the experiment results in Fig.2 of the main text. In contrast to the theory plots in Fig.2, here a combined adjustment of demodulation phase (about 20 degrees) and light polarization (ellipticity change, 4 degrees rotation on the Poincare sphere) has been done. It can be seen that the resonance on the

right has an improved match between theory and experiment, compared to that in Fig.2.

- (3) The dual-structured resonance lineshape characteristic to wall-coated cells. This alters the NMOR resonance (used for magnetometry) lineshape and makes the absolute values of optical rotations hard to compute precisely. As extensively investigated in literature, in coated cells the EIT or NMOR resonance lineshape [References: *Laser Photonics Rev.* 6, 333 (2012); *Phys. Rev. Lett.* 81, 5788 (1998)] has a narrow structure on top of a broad pedestal, where the linewidth of the latter is transit broadened and the former is determined by the multiple-pass interactions when atoms cross the laser beam multiple times with retained atomic coherence upon wall collisions. The pedestal level is determined by the random trajectories of all the atoms (which is unknown due to the unknown distribution of the coating on the wall) and can be only semi-quantitatively described with simplified model or Monte-Carlo simulations [References: *Nat. Phys.* 12, 1139 (2016); *Chin. Phys. B* 22, 033202 (2013)]. Also, upon each wall collision, the change in both the external (such as the velocity) and internal state (such as the ground state spin) of the atoms is hard to accurately model.

Here we show the dual-structured EIT and NMOR spectra from literature, as well as the NMOR spectra from our experiment. It can be seen that the NMOR spectra's tilted broad pedestal can change the lineshape of the narrow structure part, especially the resonance on the side which is used for magnetometry here. The NMOR spectra from our master equation simulation do not have the pedestal because the atomic motion and wall collisions are not considered.

Fig. 4. (color online) Example of the measured EIT spectrum exhibiting a dual structure. The inset shows the enlarged spectrum.

FIG. 3. Optical rotation dependence on the longitudinal magnetic field. Light intensity: $\approx 100 \mu\text{W}/\text{cm}^2$; the laser is tuned to the peak of the $F = 3$ component of NMOE, corresponding to a ≈ 150 MHz high frequency detuning from the peak of the fluorescence (Fig. 2). The solid line is a fit to the model described in the text. The inset shows a detailed scan of the near-zero B_z -field region.

Fig.R7 Dual-structured lineshapes in coated cells from literature. Left: EIT spectrum in a paraffin coated cylindrical cell with 2.5 cm in diameter and 7.5 cm in length [Ref: *Chin. Phys. B* 22, 033202 (2013)]. Right: NMOR spectrum in a paraffin coated spherical cell with diameter of 10 cm [Ref: *Phys. Rev. Lett.* 81, 5788 (1998)]. Note: for a similar coating quality, the larger the cell, the smaller the linewidth of the narrow-structure, which can be explained by a Ramsey-narrowing model [Ref: *Optics Express* 16, 14128 (2008)]. Also, the alkene wall-coating [Ref: *Phys. Rev. Lett.* 105, 070801 (2010)] used here has a better relaxation-preserving capacity than the paraffin coating.

Copyright statements: The left figure is reprinted with permission from [Xu Zhi-Xiang, Qu Wei-Zhi, Gao Ran, Hu Xin-Hua, and Xiao Yan-Hong, "Linewidth of electromagnetically induced transparency under motional averaging in a coated vapor cell", *Chin. Phys. B*, Vol. 22, No. 3 (2013) 033202. <http://dx.doi.org/10.1088/1674-1056/22/3/033202>. The right figure is reprinted with permission from [D. Budker, V. Yashchuk, and M. Zolotarev, *Phys. Rev. Lett.* 81, 5788 (1998)]. Copyright (2023) by the American Physical Society.

Fig.R8 Our measured NMOR spectra. The five peaks in the upper four-figure set are all the narrow structures, and the tilted base is part of the broad pedestal (see the lower four-figure set) which is not detected in full due to the limited current range of our low-noise current supply. There are five peaks because the laser frequency is being modulated here, and the laser sideband frequency component can form Lambda configuration with the central frequency peak, as shown in Fig.1 in the main text. In the lower four-figure set, the five narrow resonances in the middle do not show properly due to limited points in the sweeping magnetic field.

Finally, we would like to make a general remark on the test coil. Here, the coils surrounding the

atomic vapor cell are TEST coils which are used to produce the “to-be-measured” B fields and also to characterize the response of the magnetometer; They are widely used in the field of magnetometry in a similar way to us: the response of the atomic sensor to *known* magnetic fields produced by the coils is measured as a calibration, and then an *unknown* field from these coils is measured to check the efficacy and sensitivity of the magnetometer. We have added citations to some representative works from leading groups who used test coils in this way [Appl. Phys. Lett. 85, 4808 (2004); Nature 422, 596 (2003); Phys. Rev. A 65, 055403 (2002); Appl. Phys. Express 14 066002 (2021)]. In our experiment, establishing the map between the optical rotation signals and the B field parameters can be viewed as a generalized version of measuring the response of the magnetometer, which is analogous to the “slope” measurement in a single-parameter sensing problem. Thanks to the motional averaging enabled by the alkene wall-coating, the coil fields’ inhomogeneity has been averaged out in our experiment system. However, in normal vapor cells without wall coatings, to avoid the “amp meter issue” raised by the referee, one should use either a smaller vapor cell or larger test coils, so that the fields produced by the test coils, as well as by unknown targets, are all uniform across the volume of the cell. Such a practice of applying uniform field for testing should apply to magnetometers based on ANY scheme.

In conclusion, it is generally the case that, in vapor cell experiments, the master equation simulation cannot be used to establish a precise quantitative relation between the observable signals and the target parameters, due to various complicated conditions hard to characterize precisely in a real experiment, thus cannot be included precisely in a simulation. Even without any field inhomogeneities, the difference between the calculated and the measured optical rotation spectrum can still exist because of reasons listed above. Therefore, the ML and NN is needed to establish the precise map between the observable set and the parameter set, especially in a multiparameter sensing scenario with compact design where the one-to-one correspondence between a signal and a parameter (i.e., *direct encoding*) may not be available. It is precisely the innovative contribution of this work to propose that, one can use ML to establish the complex observables-parameters relation in compact multiparameter sensing, and we believe that our methodology can inspire researchers to simplify their sensing experiment by adopting *indirect encoding* with the aid of ML and NN.

In the revised manuscript, we have given a more comprehensive list of reasons that contribute to the discrepancy between the spectra measured and calculated by master equation. We have also provided information on field inhomogeneities from the ambient and the coils.

relu (ReLU): the acronym should be defined

Response: We have defined the acronym in our revised manuscript. ReLU means “rectified linear unit”.

Page 5

the NMOR signals are insensitive to the variation of ϕ , making the NN difficult to converge.
this issue is real, it is set by the physical nature of the system: the "difficulty to converge" is

actually due to information that is not contained in the acquired data. This implies that, under some geometries (B and k nearly aligned), the actual instrumental resolution is much worse than what is declared (e.g. in the 2nd column), and this is not dependent on the approach used to analyze the magnetometric signal.

Response: We agree with the referee on this point. When \vec{B} and \vec{k} are nearly aligned, our magnetometer scheme is not sensitive to the variation of ϕ . We have thus revised this sentence to “the NMOR signals are insensitive to the variation of ϕ when B is nearly aligned with k ”.

However, in the revised manuscript, we propose to use a three-pass single-beam configuration to solve this problem. We have added this proposal and numerical calculations with plots in the supplementary material. This idea can be implemented in future experiments using vapor cells with internal reflection mirrors.

The paragraph in the first half of the second column is unclear or very hard to understand. It likely requires a complete reformulation.

Response: This paragraph describes the process about how we get the sensitivity of vector magnetometer from NN. Essentially, the time traces of the optical rotation signal $(X_1(t), Y_1(t), X_2(t), Y_2(t))$ is collected at a sampling rate of 900/s and is converted into time traces of predicted parameter $(B(t), \theta(t), \varphi(t))$ via the NN. By fast-Fourier-transformation (FFT) of the predicted parameter time traces $(B(t), \theta(t), \varphi(t))$, one can obtain the sensitivity for each polar component of the magnetic field.

We have rewritten this paragraph as the following: “We continuously record the signal set of optical rotations (X_1, Y_1, X_2, Y_2) for about one minute at a sampling rate of 900 per second for each fixed \vec{B} , and the signal set at each time point is fed to the NN which then outputs the predicted parameter (B, θ, φ) . Consequently, the four time traces of the signals $(X_1(t), Y_1(t), X_2(t), Y_2(t))$ are converted into the three time traces $(B(t), \theta(t), \varphi(t))$. We then perform fast-Fourier-transform (FFT) on $B(t), \theta(t), \varphi(t)$ respectively, and obtain their sensitivities, where the frequency response of the magnetometer has also been considered and was acquired from the experiment with the aid of a NN [54] using a similar approach as described here.”

Page 6

*We applied a small AC magnetic field at 11 Hz to slightly vary (B, θ, ϕ) *

11Hz is not \ll the 16Hz bandwidth mentioned in page 5, thus one expects that the system response is no longer that measured under DC condition at the training stage: either the training should be made also under dynamic conditions, or the frequency range in which the DC training is assumed to be valid should be defined better.

Response: We thank the referee for pointing out this omission. As we have mentioned in our manuscript “We applied a small AC magnetic field at 11Hz to slightly vary (B, θ, φ) , and NN is trained in the parameter space near $B \approx 140nT \dots$ ”. Indeed, we made the training under

dynamic conditions. We have rewritten this sentence to make it clearer: “We applied a small AC magnetic field at 11Hz to slightly vary (B, θ, φ) , and NN is trained for the AC field in the parameter space near $B \approx 140\text{nT}$”.

DISCUSSIONS* -> *DISCUSSION

This section seems more a "conclusion" than a "discussion".

Quantitative assessments would be expected in a discussion: as an example, the sentence "...the neural network, and achieve good sensitivities on field amplitude..." should be made quantitative, replacing "good" with a number.

Response: We have heavily revised this section. We replaced “good” with numbers. In addition, we have added a full paragraph as general remarks on how to apply machine learning in other types of multi-parameter magnetometers and atomic sensors: “Our strategy of using machine learning to simplify the structure of vector NMOR-magnetometers can be extended to other types of atomic magnetometers, as well as multiparameter sensors in general, using the following procedure: (1) Identify a set of observables which are sensitive to the target parameters and can be simultaneously, if possible, recorded in the experiment. The rich degrees of freedom in the interrogating laser or broadly the electromagnetic field, for example the amplitude, polarization, spatial modes, frequency spectra etc., can be all used for encoding the information indirectly and compressively. (2) Stabilize the experiment system as a prerequisite for a robust map between the observable set and the parameter set. (3) Experimentally collect data within a suitable range of target parameters and perform the neural network training to build the map between signal set and parameter set. The NN structure is chosen according to the complexity level of the problem, and overfitting should be avoided. (4) Conduct real measurements using the trained NN.
”

The bandwidth of the magnetometer can be increased by broadening the resonance with higher laser power.

One would expect that after increasing the signal by means of cell heating, the spin exchange would make the resonance broader with no need of using power broadening. And noticeably this will rapidly lead to increase the signal amplitude, but not to enhance the sensitivity. As a general remark, such "technical" improvements should be made before collecting data for publication: I strongly suggest to apply NN and ML to already optimized magnetometric signals.

Response: In fact, our magnetometer results reported in the manuscript have already been optimized for our physical system, in terms of the sensitivities and balanced performance along three orthogonal directions, where the optimization parameters include the laser power, laser frequency detuning, laser modulation parameters, laser polarization, and in addition, we have used laser frequency lock and laser intensity lock to stabilize the system. As pointed out by referee #2, “The vector sensitivities reported are near the state of the art but are not notable by themselves.”

We agree with the referee that broadening the resonance by power broadening or spin

exchange may increase the signal but not the sensitivities. Currently, our magnetometer is optimized for sensitivities, not for bandwidth. In fact, at higher laser power, although the bandwidth of the magnetometer is increased, we observed worse sensitivities because the decrease in the resonance slope (due to power broadening) overwhelms the increase in the signal amplitude. Regarding cell heating, our vapor cell cannot be heated because the alkene wall-coating melts near 30 degrees. Therefore, our present experimental system is more suitable for relatively low frequency field measurements.

In the revised manuscript, we have removed the sentence “The bandwidth of the magnetometer can be increased by broadening the resonance with higher laser power” and instead we now refer to other physical systems that can have higher bandwidth (with spin exchange and high temperatures), since different systems have different strength and it is not the focus of this work to realize an all-purpose magnetometer.

In ref.7, the name Munoz requires a tilde.

Response: Thanks for pointing this out. We have corrected it in the revised manuscript.

Referee#2

The manuscript reports on an atomic vector magnetometer in a single optical beam geometry that uses an artificial neural network (NN) to map four simultaneous NMOR resonance signals, namely the in-phase and out-of-phase quadrature components at the modulated frequency and second-harmonic, to the magnetic field strength and direction with 100 fT/rt[Hz] and 100 uRad/rt[Hz] sensitivities. The authors demonstrate these results with a 140 nT field generated by a calibrated 3D coil system. This work is noteworthy because the vector information is acquired simultaneously in a single shot with a single beam geometry, which simplifies the sensor architecture compared to current vector magnetometers.

The quality of the data presented here is excellent and support the reported results. The paper is well-written, except for minor comments below, and previous work is properly cited. Together with the supplementary material, this manuscript gives all the necessary experimental and theoretical details. The vector sensitivities reported are near the state of the art but are not notable by themselves. In addition, the NMOR techniques are also standard and the NMOR signals used in this work have been historically reported by Budker's group to be useful for vector magnetometry. The main innovative step in this work is to apply a NN to properly map these NMOR signals to the magnetic field vector parameter space.

By using ML to effectively simplify the sensor's architecture, this work will surely impact the field of atomic magnetometry. For vector magnetometers a simpler architecture means the sensor is less prone to drifts and can be more compactly designed. This work may also be of interest to other disciplines outside of magnetometry that require multi-parameter sensing.

In view of the above, I recommend this paper for publication with some minor comments and suggestions below:

Response: We thank the referee for an overall positive assessment of this work.

- I suggest that the font size in the figures should be increased for better readability. This is especially true for the plot axes in Figures 2,3,4

Response: We thank the referee for pointing out this problem. We have fixed it in the revised manuscript.

- In Sec. II, second paragraph “..weights updated using [the] gradient descent algorithm...”

Response: We have added “the” in this sentence.

- The “relu” function in Sec. IV paragraph three is often written as “ReLU”.

Response: We have changed “relu” to “ReLU”.

- Often there is a missing “the” before “NN”. E.g. in last paragraph of sec. IV “... is in good agreement with those given by [the] NN method”.

Response: We have added “the” in front of all “NN” in the manuscript as needed.

- In Fig. 4 b1, I suggest writing something like dB instead of just B since it is a change of the magnetic field value that you are plotting here.

Response: In the revised manuscript, we have made the suggested change in Fig.4 b1.

- Revise grammar and wording in the sentence “The predicted vector...method” of the last paragraph of section IV.

Response: We have revised the original sentence “The predicted vector components of \vec{B} are consistent with the true values and the size of the error bars (standard deviations) indicates the sensitivity which is in good agreement with those given by NN method.”

It now reads “The predicted changes in the vector components of \vec{B} are consistent with the true values, as shown in Fig.4b where the sizes of the error bars (standard deviations) indicate the sensitivities, which agree with those given by the NN-aided noise analysis shown in Fig.4a.”

REVIEWERS' COMMENTS

Reviewer #1 (Remarks to the Author):

I have read the revised version of the manuscript entitled "Machine Learning Assisted Vector Atomic Magnetometry" by Xin Meng et al.

I confirm my opinion expressed at the first round review, about the rather specialistic content and about the scarce innovativity in terms of scientific knowledge. On the other hand, I do understand the opinion expressed by the Rev.2, which is used by Authors to demonstrate the innovativity of their work and the possible existence of a broad readership interested in the technical approaches presented in the paper. The different (opposite) conclusions are actually a rather subjective matter that seems quite related to the general editorial policy.

In this sense, if the work will be considered worth of being published thanks to the techniques proposed and presented, despite the lack of evident scientific innovativity, I would consider such a decision consistent and correct.

In such a case, I wish just to point out a few residual adjustments aimed to improve the paper quality, most of which are of editorial relevance.

- a) The title of sec.III should likely be "Experimental setup"
- b) The title of sec.V should likely be "Discussion" (singular)
- c) There are minor typos, e.g. a an anomalous font for a "(" 8 lines after Fig.4, several instances of anomalous quote symbols in the Supplemental material: the initial and final " use different fonts.
- d) The equation 2 in the supplemental material requires to be enlarged and probably split, because in the current version it is barely readable.

Dear editor

We would like to thank you and the referees for the hard work on reviewing our manuscript, and for the support of publishing this work. We have made all the necessary changes in the manuscript, including the editorial ones and those suggested by the referee.

Sincerely,
Yanhong Xiao

-----Below please find our point-by-point reply (in blue) to the referee report-----
Reviewer #1 (Remarks to the Author):

I have read the revised version of the manuscript entitled "Machine Learning Assisted Vector Atomic Magnetometry" by Xin Meng et al.

I confirm my opinion expressed at the first round review, about the rather specialistic content and about the scarce innovativity in terms of scientific knowledge. On the other hand, I do understand the opinion expressed by the Rev.2, which is used by Authors to demonstrate the innovativity of their work and the possible existence of a broad readership interested in the technical approaches presented in the paper. The different (opposite) conclusions are actually a rather subjective matter that seems quite related to the general editorial policy. In this sense, if the work will be considered worth of being published thanks to the techniques proposed and presented, despite the lack of evident scientific innovativity, I would consider such a decision consistent and correct.

In such a case, I wish just to point out a few residual adjustments aimed to improve the paper quality, most of which are of editorial relevance.

Reply: We thank the referee for supporting publishing the work. The presentation of the paper has been greatly improved thanks to all the comments by the referee.

a) The title of sec.III should likely be "Experimental setup"

Reply: The title has been changed to "Experimental setup" as suggested.

b) The title of sec.V should likely be "Discussion" (singular)

Reply: The title has been changed to "Discussion" as suggested.

c) There are minor typos, e.g. an anomalous font for a "(" 8 lines after Fig.4, several instances of anomalous quote symbols in the Supplemental material: the initial and final " use different fonts.

Reply: We thank the referee for pointing out these typos. We have fixed them.

d) The equation 2 in the supplemental material requires to be enlarged and probably split, because in the current version it is barely readable.

Reply: We thank the referee for pointing out this problem. We have rewritten this equation and it is now of normal size.